# Impact of living kidney donation on blood pressure and arterial stiffness: Systematic review and meta-analysis

**Maria Smyrli**[1,2]*, **Theodora Oikonomaki**[3], **Chrysanthi Skalioti**[2], **Eleni Kapsia**[2], **Ioannis N. Boletis**[2], **Smaragdi Marinaki**[2]

**1** Dialysis Unit, Nefroiatriki Green, Athens, Greece, **2** Department of Nephrology and Renal Transplantation, Laiko General Hospital, School of Medicine, National and Kapodistrian University of Athens, Athens, Greece **3** Nephrology Department "Antonios Billis", General Hospital of Evangelismos, Athens, Greece

* mariasmirli3002@hotmail.com

## Abstract

### Background

We aimed to conduct a systematic review and meta-analysis (PROSPERO CRD42023480478) regarding the impact of kidney donation on arterial stiffness indices such as pulse wave velocity (PWV) and augmentation index (AIx), along with its effect on blood pressure.

### Methods

We searched for publications related to kidney/renal donors, arterial stiffness, blood pressure, and cardiovascular risk, and included every study employing those terms. A p-value < 0.05 was considered statistically significant.

### Results

Twelve studies and 2059 individuals, with a mean age of $46.53 \pm 11.27$ years, were included in the analysis. Male donors constituted 40.6% of the participants, and the mean follow-up was $2.62 \pm 3.2$ years. Eleven studies indicated that systolic (SBP) and diastolic blood pressure remained stable within the first year after nephrectomy. However, as the follow-up period extended, especially beyond one year, both were increased (median difference (MD) of SBP was 2.09 [0.06, 4.12] over the first year and 7.7 [6.96, 8.44] over the 5 years of follow-up). Regarding PWV and AIx, assessed in 6 studies, no fluctuations were observed post-donation (MD of PWV was 0.1620 [−0.0423; 0.3662], and MD of AIx was 8.2265 [3.6450; 12.8080]). In addition, 11 studies revealed a decline in the estimated glomerular filtration rate (eGFR) after nephrectomy (MD −27.4960, p-value < 0.0001), though no albuminuria was observed. Lastly, BMI demonstrated negligible changes throughout the follow-up.

**Data availability statement:** All relevant data are within the manuscript and its Supporting information files

**Funding:** The author(s) received no specific funding for this work.

**Competing interests:** The authors have declared that no competing interests exist.

## Conclusion

Kidney donation is a relatively safe procedure, and despite the observed decline in eGFR, it does not per se impose further cardiovascular burden on the donors. However, the heterogeneity and the lack of data underscores the need for high-quality studies so as to elucidate the connection between arterial stiffness, blood pressure, and GFR level.

---

## 1. Introduction

Kidney donation is the epitome of altruism, a selfless act whose profound impact is directed solely to the recipient [1]. Thus, it becomes imperative to ensure optimal post-donation conditions for donors.

Kidney donation results in a reduction of renal mass by half, leading to adaptive glomerular hyperfiltration and a decrement in glomerular filtration rate (GFR) in the short-term after nephrectomy. Although kidney donors are not at higher absolute risk of developing end stage renal disease (ESRD) compared to non-donors in the long-term, there are concerns that the cardiovascular burden may be high [2,3]. Moreover, in response to the lack of available kidney grafts, the selection criteria for living kidney donation have been expanded, thus further reinforcing the aforementioned skepticism [4].

It has been hypothesized that a decrease in GFR following nephrectomy, may lead to increased arterial stiffness in this population, regardless of blood pressure levels [5]. In addition, it has been reported that kidney donors may experience accelerated progression of aortic stiffness, which is an independent risk factor for cardiovascular disease and all-cause mortality in the general population [5,6].

While unilateral nephrectomy may result in changes in the renin–angiotensin–aldosterone system and the vascular tone, evidence suggests that kidney donation may be associated with an increase in blood pressure over time, though a direct causal link to hypertension has not been definitively established [5,7]. However, increased arterial stiffness may increase systolic blood pressure, which in turn triggers a vicious cycle of further worsening of arterial stiffness [8].

The connection between kidney injury, blood pressure and arterial stiffness is not fully elucidated, raising significant questions about the cardiovascular risk and the overall well-being of kidney donors.

In order to assess the impact of kidney donation on cardiovascular burden, we conducted a systematic review and meta-analysis of studies evaluating blood pressure, as well as indices of arterial stiffness, both pre- and post-unilateral nephrectomy in adult living kidney donors.

## 2. Materials and methods

The systematic review and meta-analysis, registered under PROSPERO (CRD42023480478), were conducted in accordance with a pre-established protocol [9]. This protocol encompassed the research objective, strategy, and methods for

data extraction and statistical analysis, as specified by the authors, following the Preferred Reporting Items for Systematic Reviews and Meta-analyses (PRISMA) 2020 guideline (S1 File) [10,11].

We conducted a literature search in Medline, Embase, Cochrane, and PubMed to identify all relevant publication from January 2010 until January 2024.

Our inclusion criteria encompassed all types of studies, whether case series or prospective, that focused on the appraisal of validated indices of arterial stiffness and compared indicators of cardiovascular risk of adult kidney transplant donors pre- and post-nephrectomy. We did not include studies based on registry data. The search strategy employed specific Mesh search terms, including "Renal Transplantation donors," "Kidney Transplantation donors," "Cardiovascular Risk," "Pulse Wave Velocity," "Aortic Augmentation Index," and "Blood Pressure". Moreover, to reduce the risk of reporting bias, full-text articles, and abstracts published in PubMed-indexed and non-indexed journals, studies from trial registers and clinical study reports were eligible, regardless of language and publication status. Thus, we manually examined the references of each pertinent study for potential supplementary publications. Further searches involved a re-evaluation of abstracts and review articles.

The primary outcome measure was the effect of kidney donation on alterations in systolic and diastolic blood pressure, pulse wave velocity (PWV), and the aortic augmentation index (AIx) pre- and post-kidney donation. The primary safety outcome was the change in estimated glomerular filtration rate (eGFR). Secondary outcomes included changes in albuminuria levels and body mass index (BMI).

## 2.1. Quality assessment/data extraction

Two independent researchers (C.M.,V.F.) assessed the data quality of each study using the Newcastle-Ottawa Scale (NOS) to evaluate the appropriateness of the included works in the analysis [12]. The NOS scale involves the evaluation of studies based on eight characteristics, grouped into three main categories (participant selection, comparability of groups, exposure ascertainment), with scores ranging from 0 to 9. Studies with a total score of ≥7 were considered to be of high quality (S2 Table).

The year of publication, the country and the total number of participants were reported in every study. Additionally, details such as mean participant age, follow-up duration, study quality, and measurements of previously mentioned cardiovascular risk factors, along with BMI/eGFR, were outlined individually. No additional information was obtained beyond what was published in the articles.

Furthermore, the categorization by follow-up time incorporated Janki's study into the 5-year group due to its 5.1-year duration. The subsequent study in the subsequent group had an 11.3-year follow-up period. This choice was made with the consideration that it is more closely aligned with the second group than the third.

## 2.2. Statistical analysis

All the primary and secondary outcomes were calculated as mean± sd (standard deviation). We transformed the median and IQR (inter-quartile range) to mean and sd (VassarStats.net). In this study the pooled mean and sd was calculated by:

$$\text{Mean (pooled)} = \frac{\sum_{i=1}^{n} n_i . m_i}{\sum n_i} \quad \text{Sd (pooled)} = \frac{\sum_{i=1}^{n} \{(n_i - 1) . s_i\}}{\sum (n_i - n)}$$

With regard to placebo-controlled trials, we extracted the measurement results of the primary and secondary end-points only from the group of donors. We incorporated measurements of the donors compared to their own baseline values during the follow-up period. Both the initial and final measurements throughout the entire study duration were taken into consideration.

Heterogeneity was assessed with Cochrane's Q statistics and quantified using the I² stat. This statistic indicates the proportion of variability across studies attributed to heterogeneity. We used the Der Simonian-Laird estimator for tau². Meta-regression was employed to evaluate the impact of various follow-up durations and ages on both the primary and secondary outcomes. In cases of observed heterogeneity, the random effects model was utilized.

Sensitivity analysis was performed in some cases, by repeating the primary meta-analysis substituting alternative decisions or ranges of values for decisions that were arbitrary or unclear.

We assessed publication bias through the Begg-Mazumdar test and utilized the trim-and-fil method where appropriate so as to nullify the estimated bias. Additionally, we did not impute or analyze missing data.

Results were considered statistically significant if the p-value was < 0.05. All analyses were done with R-studio (version 4.3.2)

## 3. Results

Out of 467 references screened, 12 studies (11 prospective and 1 cross-sectional study) were included in the final analysis [5,13–23] (Fig 1). Trials that incorporated measurements at predefined intervals during their follow-up were categorized as trials with both initial and final measurements for each variable of interest at the conclusion of the follow-up period.

All eligible studies compared kidney donors pre- and post-nephrectomy. Studies comparing donors to other populations (i.e., recipients, healthy non-donors) were excluded. In addition, we have excluded studies that did not provide sufficient data on blood pressure values or included supplementary materials that we could not access.

In total, 2059 individuals were included in the analysis, with a mean age of 46.53 ± 11.27 years, which was similar between the different studies. Male donors constituted 40.6% of the participants in the utilized studies. All participants included in the analysis had undergone kidney donation surgery at least 6 months before enrollment and were in a stable clinical condition. In the majority of the studies, eGFR was assessed using CKD-EPI, while PWV was measured with SphygmoCor. Additionally, blood pressure (BP) measurements were conducted manually in most of the studies. The mean follow-up across all studies included in our analysis was 2.62 ± 3.2 years (median/IQR: 1 [1,4]). (S2 Table)

### 3.1. Cardiovascular risk outcomes

**3.1.1. Systolic blood pressure.** The difference in Systolic BP (SBP) after kidney donation was reported in 11 studies [5,13–15,17–23]. The range of difference between the mean SBP values at the end and at the study entry (mean difference, MD) was found to be between −0.3171 and 3.6482, with a 95% confidence interval. The common MD estimator with a random effect model is 1.6655 (Fig 2).

To elucidate the significant heterogeneity observed among the studies (I² = 98%, Q-test with a p-value<0.05), we conducted a subgroup analysis. By categorizing the studies based on their follow-up duration, we divided them into three groups (Fig 3). The first group had a follow-up time of less than a year, the second group ranged from one year to five years, and the third group extended beyond five years. Following this approach, we attained homogeneity within each of these three groups (Q within groups, p-value > 0.05, I² = 0%) and observed heterogeneity between them (Q test between groups, p-value < 0.001).

At this juncture, a sensitivity test was conducted, excluding two studies with a weight of less than 1% in the common effect model, the Gokalp 2020 and the Price 2021 study (Fig 4). Subsequently, we re-performed the meta-analysis, confirming that all the studies have an equal contribution to the analysis with a total MD of 1.57 and CI [−0.63, 3.73].

To examine the variation of the SBP among these three groups, we performed a meta-regression analysis. We noticed that within every group of each different category of time, there was homogeneity (QE, p-value>0.05), and significant difference between them (QM, p-value<0.05). It appears that SBP increases over the course of follow-up. The MD of SBP is 2.09 [0.06, 4.12] over the first year of follow-up and 7.7 [6.96, 8.44] over the 5 years of follow-up. It is important to note that the third group includes only one study.

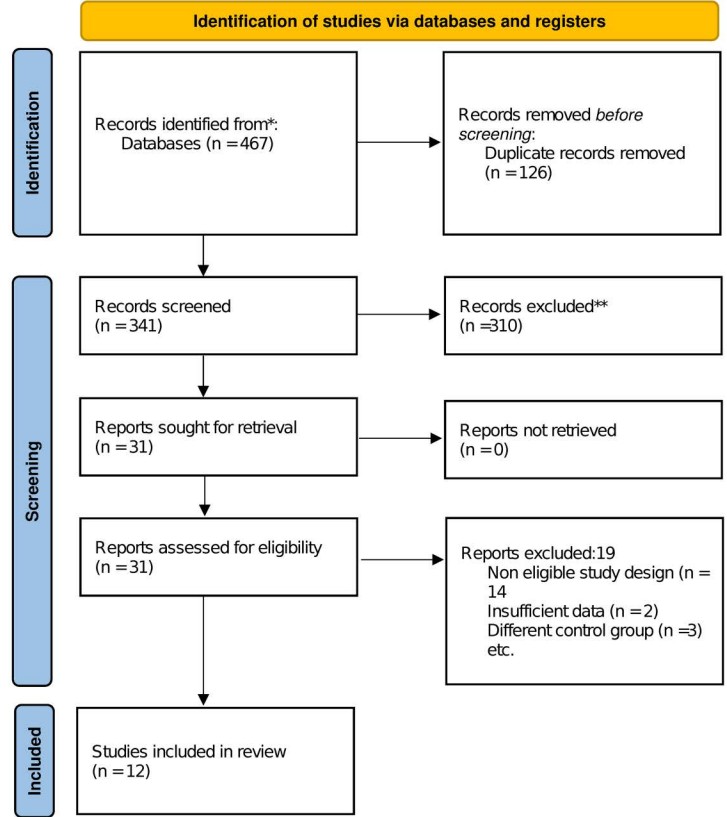

**Fig 1. Prisma 2020 flow diagram for new systematic reviews and meta-analysis.**

An identical statistical procedure was applied to age categories, yielding no significant conclusions.

The Begg-Mazumdar test, visualized through a funnel plot, indicated the presence of publication bias. To mitigate the estimated bias, the trim and fill method was employed. According to this method, the corrected MD reaffirms the previously estimated increase, but with a wider confidence interval [2.2929, 9.1703] (Fig 5).

**3.1.2. Diastolic blood pressure.** The difference in Diastolic Blood Pressure (DBP) was documented in 11 studies [5,13–15,17–23]. The range of MD in DBP observed ranged from 0.1622 to 1.9815 with a 95% confidence interval. The common MD estimator using a random-effects model is 1.0719, indicating a high degree of heterogeneity, expressed by an I² of 57.5%, and a Q-test with a p-value < 0.05 (Fig 6).

To account for this heterogeneity in the results, we applied the same methodology as the SBP analytical approach. A subgroup analysis, utilizing the three-time categories previously employed in the SBP meta-analysis, yielded consistent conclusions. Afterward, a sensitivity analysis, excluding the "Gokalp 2020" and "Price 2021" studies, pointed out that with the progression of follow-up time, the DBP tends to increase. This was confirmed with meta-regression. Specifically, there was no difference in DBP up to the first year of follow-up. However, in the subsequent follow-up period (1–5 years), a

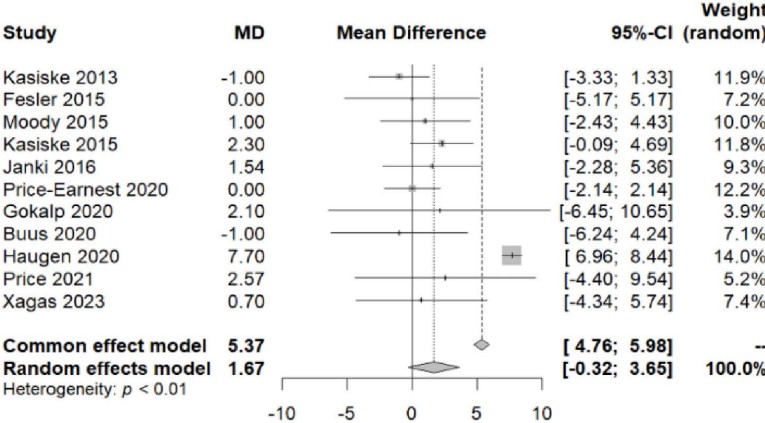

**Fig 2. Forest plot of the variation in systolic blood pressure throughout the follow-up period.**

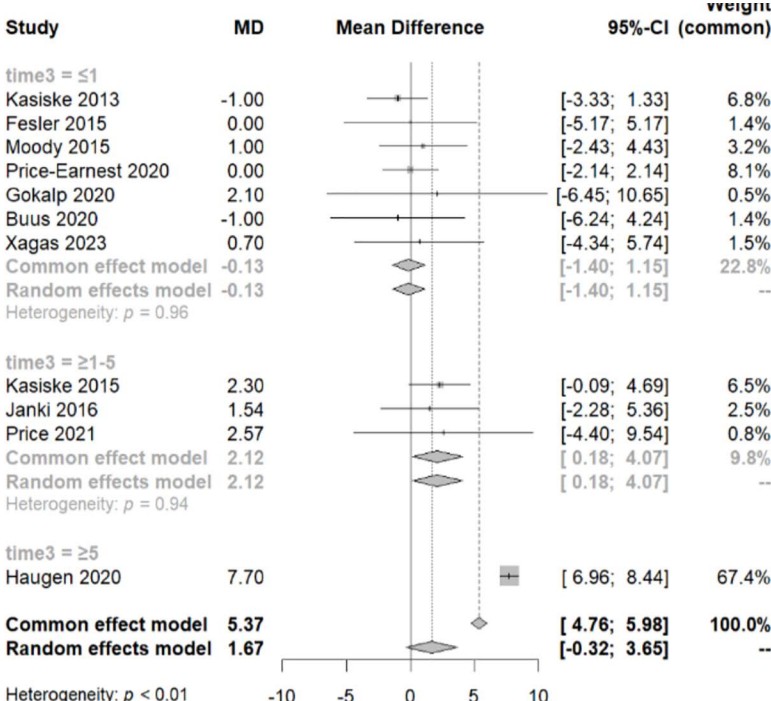

**Fig 3. Forest plot illustrating the subgroup analysis of systolic blood pressure variation across distinct follow-up time periods.** i) =<1 year, ii) 1-5 years and iii) over 5 years.

minimal but significant increase was observed, with MD of 1.35 [0.12, 1.57]. Over the 5-year period, there was a further increase, with a MD of 2.8 [2.07, 3.53] (Fig 7).

Due to the observed bias in the data, the trim and fill test suggested a broader confidence interval [0.9379; 3.0500] for the anticipated increase in DBP during the follow-up (Fig 8).

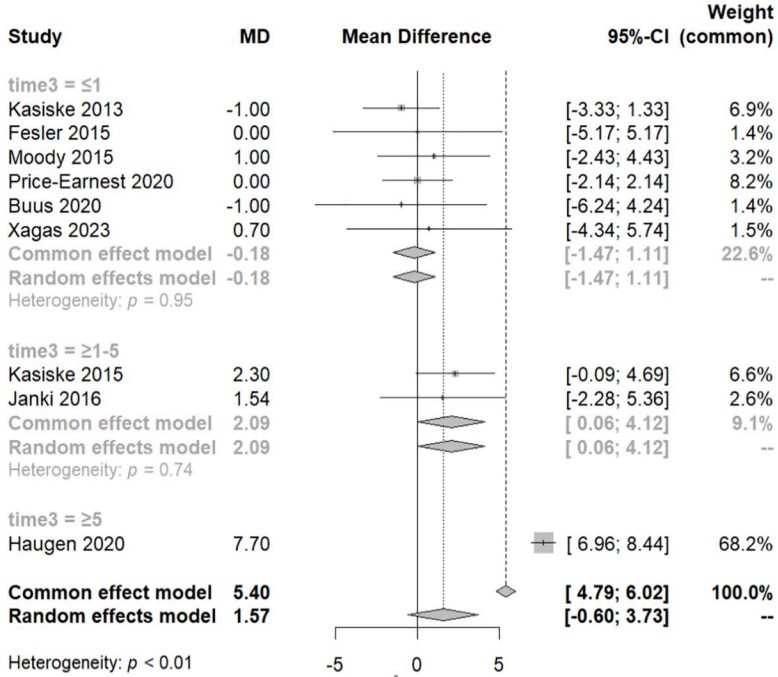

**Fig 4. Forest plot of the sensitivity analysis of the subgroups of systolic blood pressure variation across distinct follow-up time periods.**

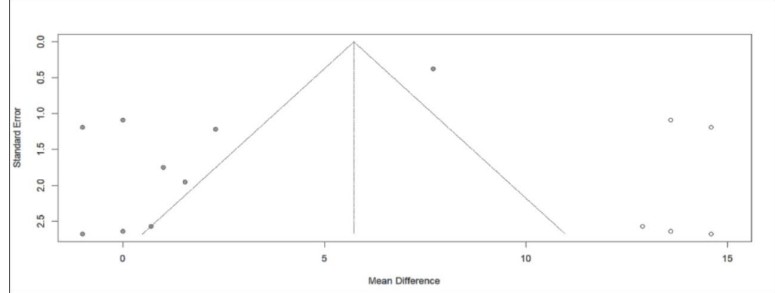

**Fig 5. Funnel plot of the systolic blood pressure data, after the trim and fill analysis.**

**3.1.3. Pulse wave velocity.** The difference in PWV, and specific in carotid-femoral PWV, was measured and reported in 6 studies [5,15,16,18,19,23]. The data exhibited homogeneity, as confirmed by the Q test (p-value > 0.05) and I² = 38%. No significant increase of PWV was observed during follow-up, as indicated by a MD of 0.1620 [−0.0423; 0.3662] using the common effect model (Fig 9).

No bias was observed among the data, as indicated by the Begg-Mazumdar test (Fig 10).

**3.1.4. Aortic augmentation index.** The AIx has been assessed in six different studies [5,15,16,18,19,23]. As heterogeneity was detected in the data (Q-test/p-value<0.05), preference was given to the random-effect model. The identified MD in AIx was 4.4242 [0.4309; 8.4174]. As per the outcomes derived from this model, there has been a reported rise in the measurements of the AIx observed at the end of the follow-up period (Fig 11). After the application of bias control measures, we acknowledged the need for correction with the trim and fill test (Fig 12). Trim and fill test suggested

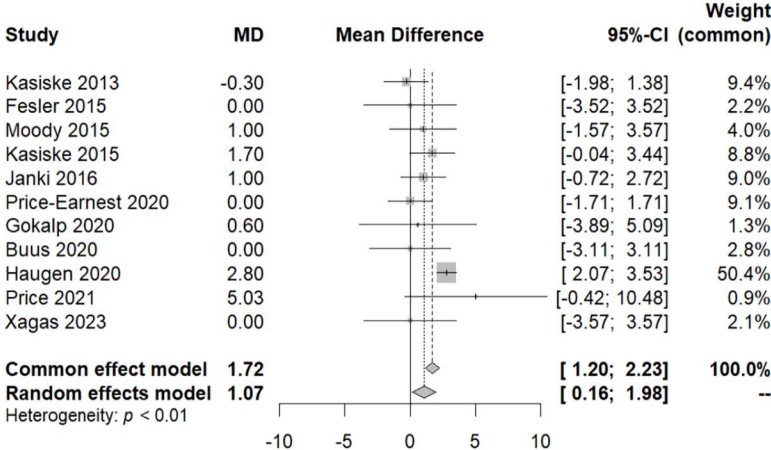

**Fig 6. Forest plot of the variation in diastolic blood pressure throughout the follow-up period.**

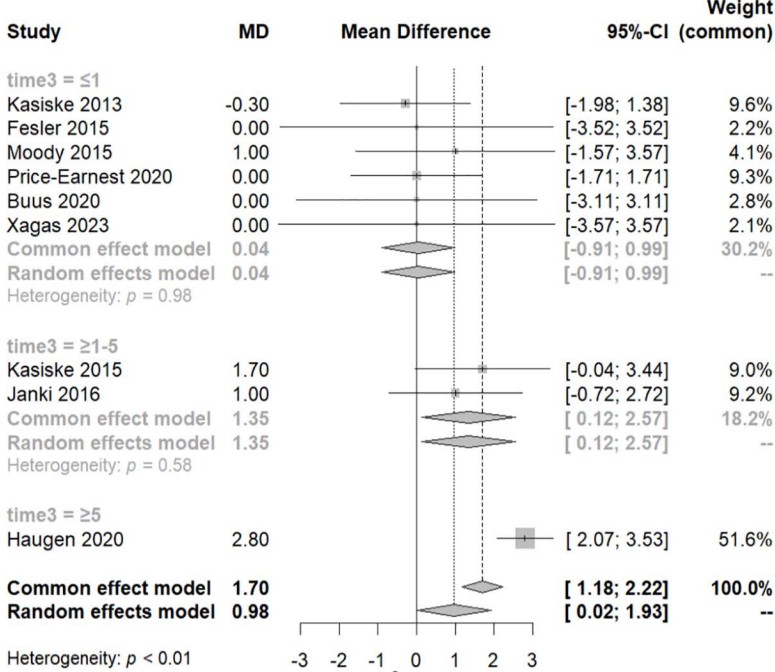

**Fig 7. Forest plot of the sensitivity analysis of the subgroups of diastolic blood pressure (DBP) variation across distinct follow-up time periods.**

a wider confidence interval 8.2265 [3.6450; 12.8080], with a prediction interval of a non-significant change in the AIx [−7.9544; 24.4074]. However, it is imperative to conduct further studies to establish more conclusive and reliable results.

**3.1.5. Secondary cardiovascular risk outcomes.** Six studies evaluated the central blood pressure (cBP) of the donors, revealing no notable variations.

**3.1.5.1 Albumin creatinine ratio**

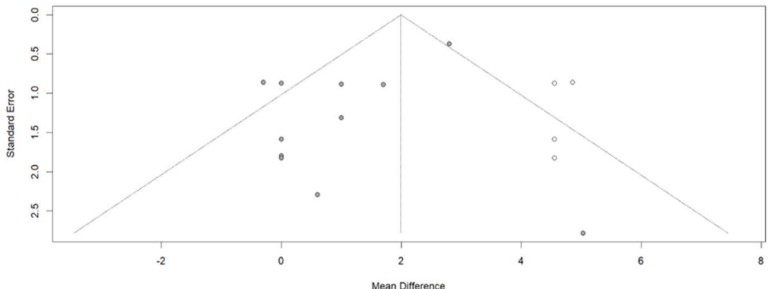

**Fig 8. Funnel plot of the DBP data, after the trim and fill analysis.**

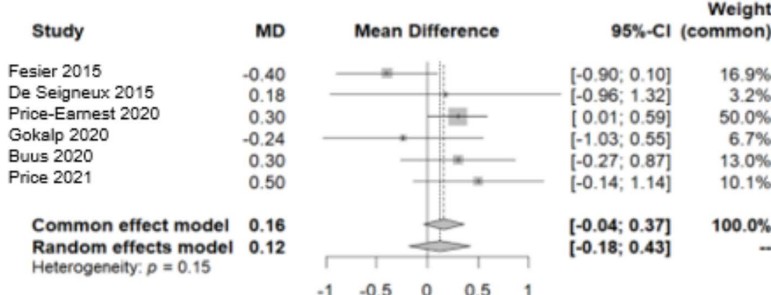

**Fig 9. Forest plot of the variation in Pulse Wave Velocity throughout the follow-up period.**

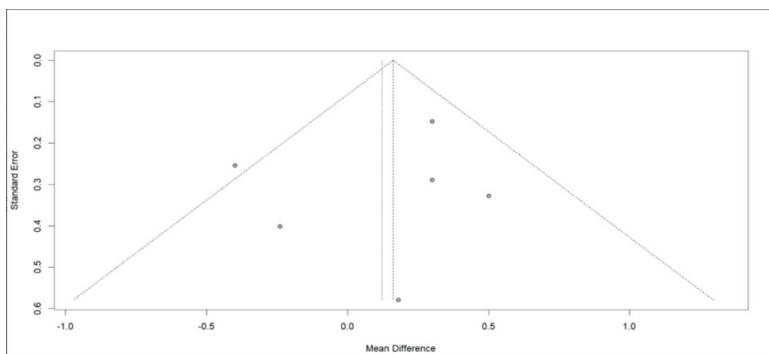

**Fig 10. Funnel plot of the Pulse Wave Velocity data.**

Limited information is available on urine albumin measurements (Albumin Creatinine Ratio, ACR), from six studies [5,15,16,18,19,23]. The data exhibited heterogeneity, and indications suggest that there was no observed change in ACR at the completion of the follow-up period (Fig 13).

### 3.1.5.2 BMI

Eight studies provided BMI data for donors at the study's outset and completion [5,13–15,17–20,23]. The measurements across these diverse studies demonstrated homogeneity ($I^2$ 7.2%, Q-test/p-value>0.05), leading to the adoption of a common effect model. Initial findings suggested a subtle yet statistically significant increase in BMI (0.4126 [0.0150; 0.8102]/ p-value: 0.042) (Fig 14).

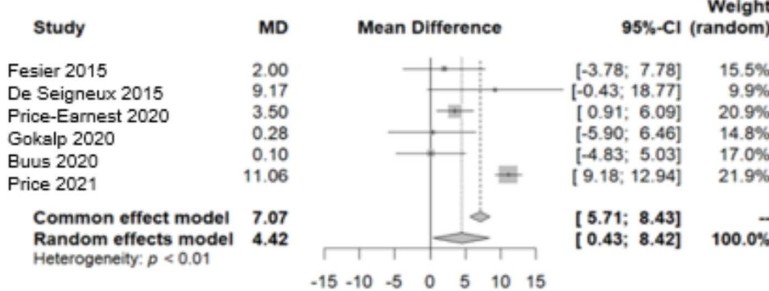

**Fig 11. Forest plot of the variation in Aortic Augmentation Index throughout the follow-up period.**

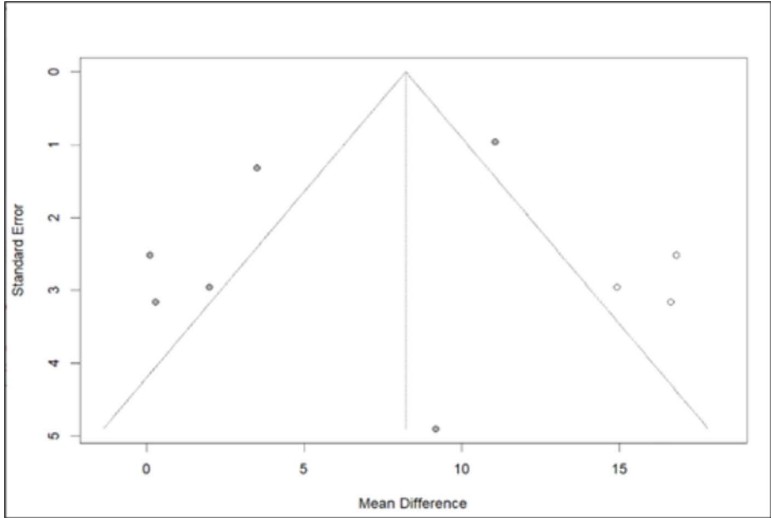

**Fig 12. Funnel plot of the Aortic Augmentation Index data after the trim and fill test.**

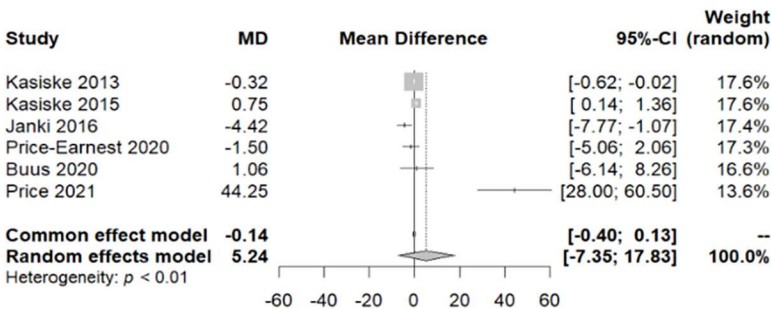

**Fig 13. Forest plot of the variation in the Albumin Creatinine Ratio throughout the follow-up period.**

Among the eight studies included, two made a marginal contribution to the conclusions, each carrying a weight of less than 2.2% (Gokalp 2020 with a weight of 1.9%, and Price 2021 with a weight of 2.2%). This prompted the implementation of a sensitivity test in which these two studies were excluded. The outcome of the sensitivity test indicated that there was no significant change in BMI at the end of the follow-up period (CI [−0.0808; 0.7314]/ p-value: 0.1164) (Fig 15).

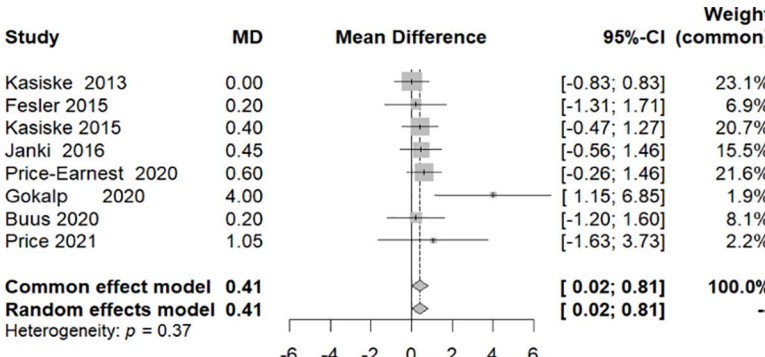

**Fig 14. Forest plot of the variation in the Body Mass Index throughout the follow-up period.**

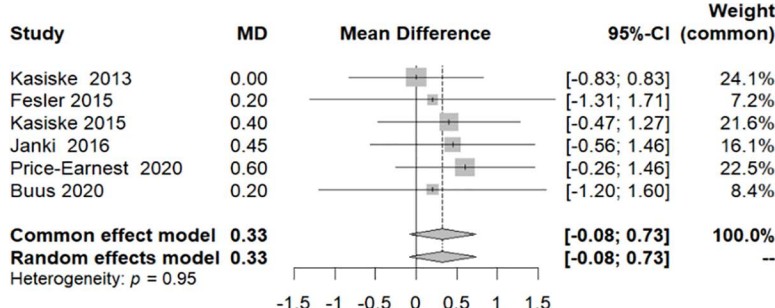

**Fig 15. Forest plot of the sensitivity analysis of the variation in the Body Mass Index throughout the follow-up period.**

The data utilized, following appropriate controls, showed no biases (Fig 16).

### 3.2. Safety outcome

No significant donor cardiovascular events were documented in any of the studies referenced during the follow-up period.

**3.2.1. eGFR.** Out of the 11 studies, a sample heterogeneity was identified with 4038 measurements, indicated by a Q value and a p-value of 0.001 [5,13–19,21–23]. Consequently, the Random Effects model was chosen, revealing a statistically significant mean GFR reduction of MD −27.4960 (p-value < 0.0001) over an average follow-up duration of 2.76 years (Fig 17).

A funnel plot was created, revealing substantial bias due to heterogeneity. Subsequently, the trim-and-fill method was employed, where, once again, in the adjusted model, a reduction in eGFR was observed with MD −21.9793 [−28.8770; −15.0817]. This reduction was smaller than that in the initial model, with a prediction indicating a general model without statistically significant change (Fig 18).

## 4. Discussion

The current systematic review and meta-analysis of 12 studies including 2059 individuals, has demonstrated that kidney donation is associated with an increase in blood pressure levels after the first year of nephrectomy, but seems to have no impact on indices of arterial stiffness. Additionally, donor kidney function, estimated by eGFR, decreased after nephrectomy, while there was no effect on albuminuria. A small but statistically significant rise of BMI was recorded among

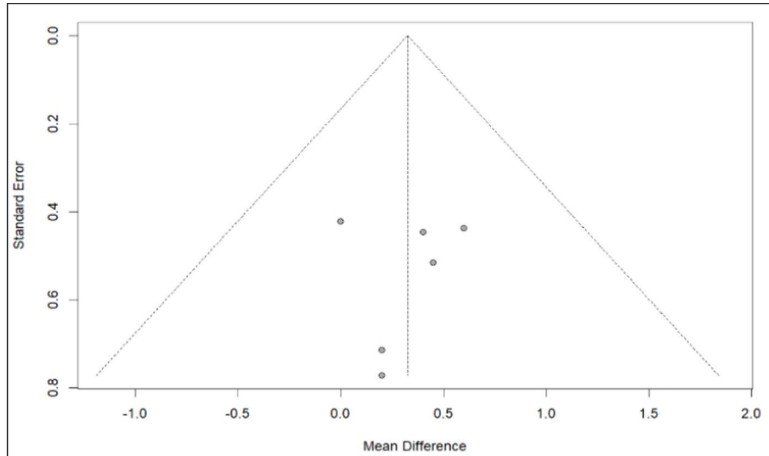

**Fig 16. Funnel plot of the Body Mass Index data.**

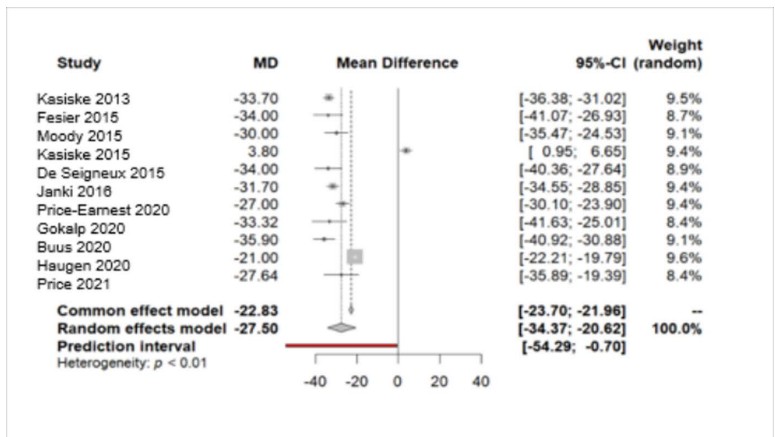

**Fig 17. Forest plot of the variation in the estimated Glomerular Filtration Rate throughout the follow-up period.**

subjects, post-donation. However, the available data are too limited to draw valid conclusions regarding alterations in the hemodynamics of kidney donors.

With regards to BP, the heterogeneity in the 11 studies involved, prompted us to analyze the evidence based on different follow up time intervals (<1 year, 1–5 years, > 5years) in order to evaluate the impact of kidney donation on blood pressure. It is important to emphasize that six studies make use of Mobil-o-graph for measurements, while the remaining five relied on office blood pressure. Our analysis showed that SBP remained stable throughout the first year following nephrectomy. According to Xagas et al. kidney donors did not experience any changes in either ambulatory BP levels or the dipping profile during the first year after nephrectomy [20]. Notably, as the follow-up period extends, discrepancies become more pronounced in systolic blood pressure, as observed in the study conducted by Haugen [21], with kidney donors experiencing gradual increases in SBP over time. Likewise, studies surpassing follow-up of one year disclosed elevated DBP values. In accordance with our results, a 2006 meta-analysis, on the risk of hypertension among kidney donors, demonstrated an increase in BP 5–10 years post-donation greater than expected with normal aging [24]. On the

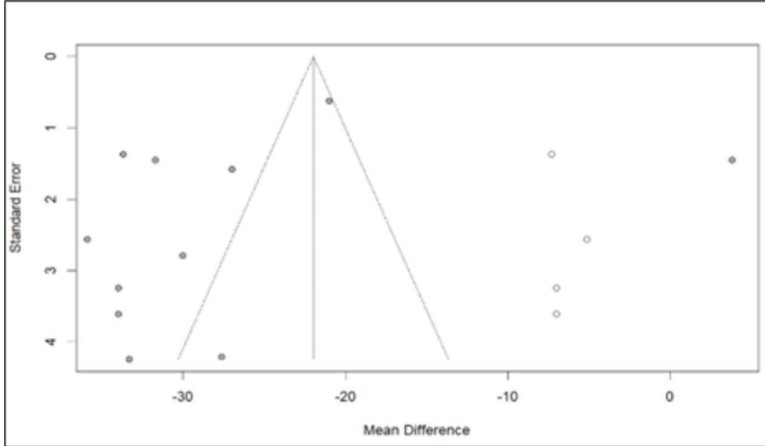

**Fig 18. Funnel plot of the estimated Glomerular Filtration Rate data after the trim and fill test.**

other hand, a more recent meta-analysis reported that SBP after nephrectomy is not higher compared to non-donor populations [2].

As for cBP, which is a better predictor of cardiovascular events, it was assessed in 6 studies but no significant fluctuations were observed between the pre- and post- donation period [6].

In assessing the impact on PWV and AIx after donation, data from six studies were analysed, all utilizing the Sphygmo-Cor device. According to the European Society of Hypertension, PWV, is considered to be a robust, non-invasive indicator in the study of arterial stiffness that reflects the potential target organ damage [25]. Our research revealed no noticeable fluctuations in PWV during the period following donation, particularly within the first year. De Seigneux et al. showed that the markers of vascular stiffness (PWV, AIx and resistive renal index) remained unchanged 1-year after donation and that calcification propensity was not affected by the loss in eGFR [16]. Similarly, Fesler et al. demonstrated no change in PWV post-nephrectomy and intriguingly its values showed a strong correlation with the pre-donation values [15]. According to a large, multicentre, prospective study, the EARNEST study, that followed 168 kidney donors and 138 matched healthy controls for 12 months, no difference was observed in PWV between the two groups [5]. However, the EARNEST study had several limitations, including a significant attrition rate, a young population and a relatively short follow-up time making it unlikely to detect significant hemodynamic or vascular alterations. On the other hand, in a small sample of 50 living kidney donors, Buus et al. noted a slight increase in PWV by 0.3m/s during follow-up [19]. As for the AIx, our analysis revealed an elevation after donation. However, the heterogeneity in the studies examined and the scarcity of data prevented us from deriving a safe and valid conclusion. Therefore, we applied bias control measures and the aforementioned increase was mitigated.

With regards to ACR, no discrepancies were noted during the follow-up in our studies. Despite Price's findings reporting mild albuminuria after nephrectomy, the remaining five studies reported no change, although it is important to acknowledge the limitations posed by the available data [23]. Of note, Janki et al, demonstrated that kidney donors, even those with eGFR<60 ml/min/1.73m$^2$, did not develop albuminuria 5 years after donation [17].

Regarding BMI, data were collected from 7 studies, revealing that no variations were identified post-donation. While both hypertension and obesity (BMI>30 kg/m$^2$) are not absolute contradictions to donation, they remain risk factors for cardiovascular events and progression of CKD [26]. Therefore, living kidney donors should be encouraged to lose weight prior to donation, especially when BMI exceeds 35 Kg/m$^2$ [27]. According to the KDIGO guidelines BP should be well-managed, typically with 1–2 anti-hypertensives and without lesions in end-organ target [27].

As for eGFR, our analysis encompasses data from 11 studies. The collective evidence indicates a reduction in eGFR over the course of follow-up. Despite the initial suggestion of increased renal function in the Kasiske study [14], further analysis, corroborated by the rest of the studies, confirms a decrease, which is greater than previously observed in preceding meta-analyses [28,29]. The magnitude of eGFR decline is consistent with the findings reported by O'Keeffe et al. in the aforementioned 2018 meta-analysis [2]. Fesler et al., showed that even though kidney donors lose half of their renal mass, they eventually experience a 32% reduction in renal function compared to the pre-donation state, which is possibly attributable to adaptive glomerular hyperfiltration [15]. Furthermore, long-term studies such as the 5-year follow-up cross-sectional study by Janki et al. indicated that the perioperative decrease in eGFR stabilizes post-donation without any further progressive decline and final values were correlated with older age, male gender, and pre-donation values [17].

Our meta-analysis has strengths and limitations. To our knowledge this is the first study that evaluates quantitatively, existing evidence on the evolution of arterial stiffness in living kidney donors. Furthermore, we included more recent studies, in an effort to provide more generalizable results relevant to the contemporary donor population. However, our review includes a small number of studies, mostly observational with small groups of participants of relatively young age and short follow-up periods. Despite our effort to minimize potential methodological problems, it is possible we failed to detect small changes in secondary end points. Unfortunately, there is a lack of large prospective, multicentre studies in the literature that would allow us to derive precise inferences on the long-term outcomes of kidney donors.

In conclusion, the current systematic review and meta-analysis demonstrated that kidney donation does not impose a further cardiovascular burden on the donor despite the decrease in eGFR, at least in the short and medium term. Kidney donors undergo a thorough mental and physical assessment prior to donation which places them in the category of the healthy population. Factors such as blood pressure, BMI and arterial stiffness which are strongly associated not only with cardiovascular risk but also with progression to renal disease, demonstrate negligible fluctuations post-donation that hardly influence the donors even 5 years after nephrectomy. However, these findings should be interpreted with caution because of the quality of the existing evidence.

Kidney donors have undertaken the ultimate act of altruism and deserve our medical and ethical commitment to document potential health risks and ensure optimal short- and long-term post-donation care with ongoing monitoring. Therefore, well-designed, methodologically sound, prospective studies of long duration, over 10 years, are of paramount importance.

## Supporting information

**S1 File. PRISMA Checklist.**
(PDF)

**S2 Table. Assessment of the quality of studies included in the review was conducted according to the Newcastle-Ottawa Scale (NOS).**
(PDF)

**S3 Table. Basic characteristics of the trials included in the study.** p = prospective, cs = case series.
(PDF)

**S4 Table. Numbered table of all studies identified in the literature search, including those that were excluded from the analyses.**
(XLSX)

## Acknowledgments

We thank the medical staff of Laiko General Hospital for their valuable work and assistance in carrying out this systematic review and meta-analysis.

## Author contributions

**Conceptualization:** Smaragdi Marinaki.

**Data curation:** Theodora Oikonomaki.

**Formal analysis:** Theodora Oikonomaki.

**Investigation:** M. Smyrli.

**Methodology:** Theodora Oikonomaki.

**Resources:** M. Smyrli.

**Supervision:** Smaragdi Marinaki.

**Writing – original draft:** M. Smyrli.

**Writing – review & editing:** Chrysanthi Skalioti, Eleni Kapsia, Ioannis N. Boletis, Smaragdi Marinaki.

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
