## [Decision Letter · Decision Letter 0]

6 Feb 2025

PONE-D-24-26455Impact of living kidney donation on blood pressure and arterial stiffness. Systematic review and meta-analysis.PLOS ONE

Dear Dr. Smyrli,

Thank you for submitting your manuscript to PLOS ONE. After careful consideration, we feel that it has merit but does not fully meet PLOS ONE’s publication criteria as it currently stands. Therefore, we invite you to submit a revised version of the manuscript that addresses the points raised during the review process.

We look forward to receiving your revised manuscript.

Kind regards,

Satish G Patil, PhD

Academic Editor

PLOS ONE

Journal Requirements:

2. Please ensure that your PRISMA flow diagram is included in your main manuscript file as Figure 1; please see the PLOS ONE submission guidelines for systematic reviews and meta-analyses at https://journals.plos.org/plosone/s/submission-guidelines#loc-systematic-reviews-and-meta-analyses.

3. As required by our policy on Data Availability, please ensure your manuscript or supplementary information includes the following: 

Reviewers' comments:

Reviewer's Responses to Questions

**Comments to the Author**

1. Is the manuscript technically sound, and do the data support the conclusions?

Reviewer #1: Yes

Reviewer #2: Partly

2. Has the statistical analysis been performed appropriately and rigorously? 

Reviewer #1: Yes

Reviewer #2: Yes

3. Have the authors made all data underlying the findings in their manuscript fully available?

Reviewer #1: Yes

Reviewer #2: Yes

4. Is the manuscript presented in an intelligible fashion and written in standard English?

Reviewer #1: Yes

Reviewer #2: Yes

5. Review Comments to the Author

Reviewer #1: Authors aimed to aimed to conduct a systematic review and meta-analysis regarding the

impact of kidney donation on arterial stiffness indices such as 24 pulse wave velocity (PWV)

and augmentation index (AIx), along with its effect on blood pressure.

Some results described in the abstract are not in line with the results later described along the

article. In abstract section is referred that “BMI remained unchanged throughout the follow-up”. However, in page 14, line 291 is stated that “A small but significant rise of BMI was

recorded among subjects, post-donation”.

There are also other misleading statements along the paper. In introduction it is stated that a

causal inference between kidney donation and hypertension has not been demonstrated; but,

as stated latter in discussion, reference 24 had already described and increase in BP 5-10 years

post donation greater than expected with normal aging.

This information could be confusing to the reader and may be changed accordingly.

Minor points: Page 3, line 48, what that authors mean with “sub-duplication of renal mass"

Reviewer #2: The authors analysed the results of studies examining changes in arterial stiffness indices (pulse wave velocity-PWV and augmentation index-AIx), body mass index (BMI), blood pressure and estimated glomerular filtration rate (eGFR) parameters after nephrectomy in kidney transplant donors (n: 2059, mean age 46.53±11.27 years and mean follow-up 2.62±3.2 years). The subject of the study is interesting. It is a subject worth examining regarding the increasing trend of related living donor kidney donation in most countries due to the scarcity of deceased donors.

They reported a decrease in eGFR after nephrectomy. Arterial stiffness, albuminuria and BMI did not change during follow-up. They showed that systolic and diastolic blood pressure remained stable the first year after nephrectomy, but both increased as the follow-up period increased (especially over one year).

Can the parameter changes be evaluated by distinguishing between standard and expanded criteria (marginal) donors in this population? Is there a correlation between donor age and changes in the studied parameters? The rate of new antihypertensive drug initiation or increase in drug requirement after nephrectomy in the donors in the studies can also be discussed (if data is sufficient).

Abbreviations should be reviewed.

References should be written according to the journal rules.

6. PLOS authors have the option to publish the peer review history of their article (what does this mean? ). If published, this will include your full peer review and any attached files.

**Do you want your identity to be public for this peer review?** For information about this choice, including consent withdrawal, please see our Privacy Policy .

Reviewer #1: **Yes: ** Sara Querido

Reviewer #2: **Yes: ** ALPARSLAN ERSOY

---

## [Author Response · Author response to Decision Letter 1]

15 Mar 2025

Editor’s Comments: When submitting your revision, we need you to address these additional requirements. Please ensure that your manuscript meets PLOS ONE's style requirements, including those for file naming. The PLOS ONE style templates can be found at. https://journals.plos.org/plosone/s/file?id=wjVg/PLOSOne_formatting_sample_main_body.pdf and https://journals.plos.org/plosone/s/file?id=ba62/PLOSOne_formatting_sample_title_authors_affiliations.pdf

Response to editor. Thank you for your comment. After careful consideration, we have updated the file names and changed the format of the supplementary material to PDF, as per the PLOS ONE style guidelines. We appreciate your guidance and attention to detail.

Editor’s Comments. Please ensure that your PRISMA flow diagram is included in your main manuscript file as Figure 1; please see the PLOS ONE submission guidelines for systematic reviews and meta-analyses at https://journals.plos.org/plosone/s/submission-guidelines#loc-systematic-reviews-and-meta-analyses.

Response to editor. Thank you for your comment. We have included the PRISMA flow diagram in the main manuscript as Figure 1, and we have also adjusted the numbering of all subsequent figures accordingly. This change can be found on line 139 & 142.

Editor’s Comments: As required by our policy on Data Availability, please ensure your manuscript or supplementary information includes the following: A numbered table of all studies identified in the literature search, including those that were excluded from the analyses. For every excluded study, the table should list the reason(s) for exclusion.

Response to Editor: Thank you for your comment. As requested, you will find a numbered table in the supplementary material (S4 table) that includes both the studies accepted and those excluded from the analyses, along with the reasons for their exclusion. This information is provided on line 468-469.

Editor’s Comments: If any of the included studies are unpublished, include a link (URL) to the primary source or detailed information about how the content can be accessed.

Response to Editor. Thank you for your comment. All the studies included in our analysis are published, and therefore, we did not need to provide any URLs or access information for unpublished studies. We appreciate your attention to detail.

Editor’s Comments: A table of all data extracted from the primary research sources for the systematic review and/or meta-analysis. The table must include the following information for each study: Name of data extractors and date of data extraction Confirmation that the study was eligible to be included in the review. All data extracted from each study for the reported systematic review and/or meta-analysis that would be needed to replicate your analyses.

Response to the Editor. Thank you for your comment. As requested, you will find the table containing all the data extracted from the primary research sources in the supplementary material (S3 table, line 467). This table includes the name of the data extractors, the date of data extraction, confirmation of study eligibility (assessed through NOS score), and all data extracted from each study necessary to replicate our analyses.

Editor’s Comments: If data or supporting information were obtained from another source (e.g. correspondence with the author of the original research article), please provide the source of data and dates on which the data/information were obtained by your research group.

Response to the Editor. Thank you for your comment. We only used data that were published in the studies and did not seek any further information. To clarify this, we have added the following sentence in our manuscript: "No additional information was obtained beyond what was published in the articles." (Lines 108–109)

Editor’s Comments: If applicable for your analysis, a table showing the completed risk of bias and quality/certainty assessments for each study or outcome. Please ensure this is provided for each domain or parameter assessed. For example, if you used the Cochrane risk-of-bias tool for randomized trials, provide answers to each of the signaling questions for each study. If you used GRADE to assess certainty of evidence, provide judgements about each of the quality of evidence factor. This should be provided for each outcome.

Response to editor. It was no applicable to show the complete risk of bias in a separate table for each study but we have NOS score for publication assessment (S2 table Line 489-490). As for the bias assessments, we use Begg-Mazumdar test and utilized the trim-and-fil method where appropriate so as to nullify the estimated bias. Line 131-132.

Editor’s Comments: An explanation of how missing data were handled.

Response to editor. Thank you for your comment. We did not have any missing data in our analysis. For each parameter, we included only studies with available data for the main variables of interest. To clarify this, we have added the following sentence in our manuscript: "Only studies with available data for the main variables of interest were included in the analysis." (Lines 132–133).

Reviewer #1 Comments: Authors aimed to aimed to conduct a systematic review and meta-analysis regarding the impact of kidney donation on arterial stiffness indices such as 24 pulse wave velocity (PWV) and augmentation index (AIx), along with its effect on blood pressure. Some results described in the abstract are not in line with the results later described along the article. In abstract section is referred that “BMI remained unchanged throughout the follow-up”. However, in page 14, line 291 is stated that “A small but significant rise of BMI was recorded among subjects, post-donation”.

Response to reviewer #1: Thank you for your valuable comment. You are absolutely right, and we have revised the abstract to ensure consistency with the discussion. Specifically, we have modified the sentence in the abstract to: "Lastly, BMI demonstrated negligible changes throughout the follow-up (line 37-38)”. Simultaneously, we have adjusted the discussion to maintain accuracy: "A small but statistically significant rise of BMI was recorded among subjects post-donation" (line 294).

Reviewer #1 Comments: There are also other misleading statements along the paper. In introduction it is stated that a causal inference between kidney donation and hypertension has not been demonstrated; but, as stated latter in discussion, reference 24 had already described and increase in BP 5-10 years post donation greater than expected with normal aging. This information could be confusing to the reader and may be changed accordingly.

Response to reviewer #1: Thank you for your insightful comment. We acknowledge the potential for confusion and have revised the sentence in the introduction to provide greater clarity. The updated text now reads: "While unilateral nephrectomy may result in changes in the renin–angiotensin–aldosterone system and vascular tone, evidence suggests that kidney donation may be associated with an increase in blood pressure over time, though a direct causal link to hypertension has not been definitively established." (Lines 60–64). This revision ensures consistency with the discussion section while accurately reflecting the current evidence. We appreciate your careful review and valuable feedback.

Reviewer #1 Comments: Minor points: Page 3, line 48, what that authors mean with “sub-duplication of renal mass"

Response to reviewer #1. Thank you for your comment. To improve clarity, we have revised the statement as follows: "Kidney donation results in a reduction of renal mass by half, leading to adaptive glomerular hyperfiltration and a decrement in glomerular filtration rate (GFR) in the short-term after nephrectomy " (Lines 48-49)

Reviewer #2: The authors analysed the results of studies examining changes in arterial stiffness indices (pulse wave velocity-PWV and augmentation index-AIx), body mass index (BMI), blood pressure and estimated glomerular filtration rate (eGFR) parameters after nephrectomy in kidney transplant donors (n: 2059, mean age 46.53±11.27 years and mean follow-up 2.62±3.2 years). The subject of the study is interesting. It is a subject worth examining regarding the increasing trend of related living donor kidney donation in most countries due to the scarcity of deceased donors. They reported a decrease in eGFR after nephrectomy. Arterial stiffness, albuminuria and BMI did not change during follow-up. They showed that systolic and diastolic blood pressure remained stable the first year after nephrectomy, but both increased as the follow-up period increased (especially over one year). Can the parameter changes be evaluated by distinguishing between standard and expanded criteria (marginal) donors in this population?

Response to reviewer #2. Thank you for your thoughtful comment. Unfortunately, the studies included in our analysis did not specify whether the kidney transplant donors were classified as marginal or expanded criteria donors. Therefore, we were unable to evaluate the parameter changes based on these distinctions. We acknowledge that this would be an interesting area for future research, and we suggest that future studies explore the potential differences between standard and expanded criteria donors in this population.

Reviewer #2 Comments: Is there a correlation between donor age and changes in the studied parameters?

Response to reviewer #2. Thank you for your valuable comment. In the studies included in our analysis, the age distribution of the donors was relatively similar across studies, which limited the ability to draw meaningful statistical conclusions regarding the correlation between donor age and changes in the studied parameters. As such, we were unable to assess this relationship in a way that would yield valid conclusions.

Reviewer #2 Comments: The rate of new antihypertensive drug initiation or increase in drug requirement after nephrectomy in the donors in the studies can also be discussed (if data is sufficient).

Response to reviewer #2. Thank you for this excellent suggestion. Unfortunately, the studies included in our analysis did not provide data on the rate of new antihypertensive drug initiation or the increase in drug requirements following nephrectomy. As a result, we were unable to discuss this aspect in our review.

Reviewer #2 Comments: Abbreviations should be reviewed.

Response to reviewer #2. Thank you for your comment. We have thoroughly reviewed all abbreviations in the manuscript and have made the necessary revisions. The abbreviations are now highlighted in yellow throughout the article for clarity and consistency. We appreciate your attention to detail.

Reviewer #2 Comments: References should be written according to the journal rules.

Response to reviewer #2. Thank you for your comment. We have revised the reference format to align with the journal's guidelines. Specifically, we have changed the citation style to use square brackets instead of parentheses. We appreciate your helpful suggestion.

---

## [Decision Letter · Decision Letter 1]

13 May 2025

Impact of living kidney donation on blood pressure and arterial stiffness. Systematic review and meta-analysis.

PONE-D-24-26455R1

Dear Dr. Smyrli,

We’re pleased to inform you that your manuscript has been judged scientifically suitable for publication and will be formally accepted for publication once it meets all outstanding technical requirements.

Kind regards,

Satish G Patil, PhD

Academic Editor

PLOS ONE

Additional Editor Comments (optional):

Reviewers' comments:

Reviewer's Responses to Questions

**Comments to the Author**

1. If the authors have adequately addressed your comments raised in a previous round of review and you feel that this manuscript is now acceptable for publication, you may indicate that here to bypass the “Comments to the Author” section, enter your conflict of interest statement in the “Confidential to Editor” section, and submit your "Accept" recommendation.

Reviewer #2: All comments have been addressed

2. Is the manuscript technically sound, and do the data support the conclusions?

Reviewer #2: Partly

3. Has the statistical analysis been performed appropriately and rigorously? 

Reviewer #2: I Don't Know

4. Have the authors made all data underlying the findings in their manuscript fully available?

Reviewer #2: Yes

5. Is the manuscript presented in an intelligible fashion and written in standard English?

Reviewer #2: Yes

6. Review Comments to the Author

Reviewer #2: I have no additional comments. Perhaps the limitations and future areas of work on this subject could have been written more clearly.

7. PLOS authors have the option to publish the peer review history of their article (what does this mean? ). If published, this will include your full peer review and any attached files.

**Do you want your identity to be public for this peer review?** For information about this choice, including consent withdrawal, please see our Privacy Policy .

Reviewer #2: **Yes: ** ALPARSLAN ERSOY

---

## [Editor Report · Acceptance letter]

PONE-D-24-26455R1

PLOS ONE

Dear Dr. Smyrli,

I'm pleased to inform you that your manuscript has been deemed suitable for publication in PLOS ONE. Congratulations! Your manuscript is now being handed over to our production team.

Kind regards,

on behalf of

Prof. Dr. Satish G Patil

Academic Editor

PLOS ONE